# Bayesian Binary Mixture Models as a Flexible Alternative to Cut-Off Analysis of ELISA Results, a Case Study of Seoul Orthohantavirus

**DOI:** 10.3390/v13061155

**Published:** 2021-06-16

**Authors:** Arno Swart, Miriam Maas, Ankje de Vries, Tryntsje Cuperus, Marieke Opsteegh

**Affiliations:** Centre for Infectious Disease Control, Centre for Zoonoses and Environmental Microbiology, National Institute for Public Health and the Environment (RIVM), P.O. Box 1, 3720 BA Bilthoven, The Netherlands; miriam.maas@rivm.nl (M.M.); ankje.de.vries@rivm.nl (A.d.V.); tryntsje.cuperus@rivm.nl (T.C.); marieke.opsteegh@rivm.nl (M.O.)

**Keywords:** ELISA, mixture models, serology, SEOV, seoul orthohantavirus, cut-off analysis

## Abstract

Serological assays, such as the enzyme-linked immunosorbent assay (ELISA), are popular tools for establishing the seroprevalence of various infectious diseases in humans and animals. In the ELISA, the optical density is measured and gives an indication of the antibody level. However, there is variability in optical density values for individuals that have been exposed to the pathogen of interest, as well as individuals that have not been exposed. In general, the distribution of values that can be expected for these two categories partly overlap. Often, a cut-off value is determined to decide which individuals should be considered seropositive or seronegative. However, the classical cut-off approach based on a putative threshold ignores heterogeneity in immune response in the population and is thus not the optimal solution for the analysis of serological data. A binary mixture model does include this heterogeneity, offers measures of uncertainty and the direct estimation of seroprevalence without the need for correction based on sensitivity and specificity. Furthermore, the probability of being seropositive can be estimated for individual samples, and both continuous and categorical covariates (risk-factors) can be included in the analysis. Using ELISA results from rats tested for the Seoul orthohantavirus, we compared the classical cut-off method with a binary mixture model set in a Bayesian framework. We show that it performs similarly or better than cut-off methods, by comparing with real-time quantitative polymerase chain reaction (RT-qPCR) results. We therefore recommend binary mixture models as an analysis tool over classical cut-off methods. An example code is included to facilitate the practical use of binary mixture models in everyday practice.

## 1. Introduction

In human and veterinary science, serological assays are important tools for establishing the seroprevalence of various pathogens in a population. Such tests are typically low-cost, high-throughput, and can be executed rapidly. Serological assays can be used to assess antibody levels in serum (or similar fluid host material). In cases where enzyme-linked immunosorbent assay (ELISA) is applied for this purpose, the antibody concentration is usually reflected by chromogenic changes, measured as the optical density (OD). There is a clear intuition that ‘high’ OD levels should correspond with (previous) infection, and ‘low’ OD levels with the absence of antibodies specific to the pathogen of interest. However, both seropositive and seronegative individuals show heterogeneity in their response: seronegative ones due to variation in cross-reacting antibodies and variation in general antibody response. Seropositive individuals vary due to different infection doses, individual variation in antibody response and waning response over time [1].

A typical seroprevalence study would include sera from a sample of the target population and classify individuals according to their individual measures of antibody level. For this classification, often a ‘cut-off value’ is chosen, below which an individual is assumed to be seronegative. The determination of a suitable cut-off is a sensitive issue. Roughly in order of increasing complexity, the following approaches are found in practice [1]:

*Using a known negative population.* With access to a known negative population, the sample mean $\bar{X}$ plus a multiple of the standard deviation *S*, may be used as a cut-off. This assumes that the distribution of OD values in the negative population is Gaussian, implying that $\bar{X}+3S$ approximates the 99.7th quantile, resulting in the false positive rate to below 0.3%. However, this assumption rarely holds. Rather, OD values are usually normally distributed on the log-scale [1,2,3]. Moreover, the sensitivity remains unknown, and in the absence of an independent estimate of sensitivity, the prevalence cannot be reliably estimated.

*Using a ‘golden standard’.* The cut-off can be chosen in such a way as to maximize concordance between another test and the results of the serological assay. This is somewhat unsatisfactory since it presumes one test (i.e., ‘the golden standard’) to be absolutely superior over the other.

*Using both a known negative and a known positive population.* Having access to an additional known positive population allows for estimating both a sensitivity and specificity at any cut-off. An ROC/AUC (receiver-operator Characteristic/Area Under the Curve) analysis can then aid the researcher in selecting a cut-off with favorable properties [4], and also enables a correction of the prevalence using the Rogan–Gladen estimator [5]. However, it is unsatisfactory to obtain a corrected prevalence that no longer matches the prevalence based on the individual classifications.

Moreover, for all of these methods, individuals with a predefined infection status may show less variability in OD value than individuals from the target population, e.g., when experimentally infected animals are used for the known positive population, or animals raised under SPF (specific-pathogen-free) conditions are used as a known negative population.

In practice, often a limited number of negative control sera, sometimes only those added to each plate to correct for plate-to-plate variation, are used to estimate a cut-off value. However, large numbers of reference sera are needed to reliably estimate a mean and standard deviation. In Jacobson [1], calculations are performed, suggesting hundreds of infected and uninfected reference animals are needed to obtain reasonable accuracy. Furthermore, caution must be taken, and cut-off values based on controls repeated over plates should be avoided, since they represent the uncertainty of the OD values of the controls, not the variation within a population.

The techniques outlined above also do not offer (1) measures of uncertainty of the results; (2) extensions to more complicated settings (e.g., spatial variation, or risk-factors such as age); and (3) extensions to more than two infectious states (e.g., positive, negative, or infected with a related strain). Clearly, the often used cut-off value for classifying ELISA results has some limitations.

The need for the proper inclusion of test accuracy characteristics, and the other pitfalls involved, was pointed out before [2,6]. In the latter publication, latent class models set in a frequentist or Bayesian framework were advocated. The mixture models that we propose in the current work can also be viewed as such.

In a binary mixture model, the population is described by a mixture of two components, in which there is not a sharp cut-off, but rather a twilight zone of uncertain classification [7]. The statistical challenge is to capture those components, and also characterize the uncertainty. The goal of this paper is to introduce a more appropriate technique to analyze ELISA results, built around the concept of ‘binary mixture modeling’. This class of models has properties that a cut-off model lacks: direct estimation of prevalence, quantification of uncertainty in the estimates, and extension to complicated settings (in this paper, the sample matrix: serum or heart fluid). A further advantage is the proper treatment of censored values (measurements outside of the domain where the assay operates). Mixture models and the Bayesian treatment of the subject have been reported before, however, to our knowledge, a user-friendly model including plate-to-plate variation and the inclusion of covariates has not been presented before. To show how binary mixture models can be used in serological studies, we used the model on the example of Seoul orthohantavirus (SEOV) in rats. This rat-borne virus has a worldwide distribution and has gained renewed interest in the last decade, when human cases of hemorrhagic fever with renal syndrome (HFRS) were diagnosed in Europe and the USA [8]. We compared the model with classical cut-off methods and independent real-time quantitative polymerase chain reaction (RT-qPCR) results and show that the binary mixture model offers various properties that make it a better alternative to classical cut-off methods.

## 2. Materials and Methods

### 2.1. Data

Data from three different sets of brown rats were used for this study. The main group comprised pet and feeder rats which were tested for SEOV in 2018 [9]. Rats from different origins were tested post mortem to determine the spread of SEOV in the Dutch captive rat populations. The second group of rats consisted of wild rats that were captured in 2018 at four different locations throughout the Netherlands for a study on leptospirosis in rats and that were opportunistically tested for SEOV. Rats of both groups were stored at −20 ${}^{\circ }$C until dissection, when heart fluid was collected to test for SEOV seropositivity. Both studies were approved by the Dutch Central Animal Experiments Committee (project number AVD3260020172104). The third group of rats comprised 76 feeder rats that were collected in 2016, after two patients with SEOV were diagnosed [10]. Fifty-six rats came from one breeding farm where one patient worked, whereas the other 10 were feeder rats owned by the patient. Ten feeder rats came from the breeding farm where patient 2 worked. From five of these rats of patient one, heart fluid was collected post mortem. From the other 71 animals, serum was collected. Of all rats, lung tissue was collected for SEOV RNA detection by RT-qPCR, as described before [11]. In the current analysis, we removed seven rats: the rats that were tested on heart fluid in the feeder rat study since without these rats the modeling was greatly simplified because the ‘study’ (the study that the rats originated from) and ‘sample matrix’ (the material that the test was performed on, hearth fluid or serum) then coincide. We also removed rats that were included as plate controls. The final data are presented in Table 1.

For serology, in all three studies, optical density values were obtained using ELISA (Hantavirus Dobrava/Hantaan IgG Elisa; Progen Biotechnik GmbH, Heidelberg, Germany), with rabbit-α-rat IgG horseradish peroxidase-labeled IgG (Sigma-Aldrich Chemie B.V., Zwijndrecht, The Netherlands) as a conjugate, to enable the detection of anti-SEOV antibody levels in rats as described by [9]. OD values were measured at 450 nm and corrected by reference measurement at 650 nm. On each plate, a panel of rat sera was included as controls: negative controls, positive controls, and some samples with intermediate OD values.

### 2.2. Plate-to-Plate Variation

It is a common phenomenon that entire ELISA plates have higher or lower OD values than the average. Each plate contains a number of controls, that ideally should not vary over plates. We may use these controls to derive corrections for each plate, where we follow the approach introduced in [12]. Briefly, for each control, all OD values were log10-transformed, the average over all plates was calculated, and for each plate, a linear model was set up with the averages of the controls as outcome (y axis), and the log-transformed OD value on the current plate as predictor (x axis). Using this linear model to correct all values on the plate will push all values closer to the average and hence reduce variation over plates.

### 2.3. Cut-Off Analysis

Cut-off values were defined based on the wild rat dataset, which only contained RT-qPCR negative animals. Four cut-off values were defined: the mean plus two or three standard deviations, in both linear and log10-scale. In the following, all OD values and cut-off values were reported in log10-scale.

### 2.4. Mixture Modeling

In this section, we will outline the concepts behind the binary mixture models which we propose as an alternative to cut-off-based methods. The assumptions behind the mixture model are the following:The distribution of logOD values in both the positive and negative populations is Gaussian;The two components of the mixture each represent natural variation between animals.

When the distribution of the entire population is viewed as a weighted sum of the two populations, then the coefficient of the positive component (i.e., the mixing parameter) is exactly the prevalence. For individual rats, the model does not assume a rigid decision boundary, but gives probabilities of positivity or negativity based on the two components of the mixture. In binary mixture models, multiple datasets can be included. We included each of the data sets of Table 1. Any relevant measurement strengthens confidence in the mathematical location and the spread of the distribution for each component. Of course, the prevalence in each study could differ, therefore this is included in the model description. A convenient way of describing the data and the model was to adopt a Bayesian framework. In this model, the OD values for negative animals are normally distributed on the log10-scale, with a probability density function $f(x;\mu^{-},\sigma^{-}|\mathrm{neg})$, meaning ’the likelihood of the logOD value distribution at data point *x*, given that the animal is negative’. The mean is μ, standard deviation σ and the minus superscript denotes seronegativity. The positive component is defined as analogous. The expression for a binary mixture model can be obtained by conditioning on the possible outcomes:$$f\left(x\right)=f\left(x;\mu^{-},\sigma^{-}|\mathrm{neg}\right)p\left(\mathrm{neg}\right)+f\left(x;\mu^{+},\sigma^{+}|\mathrm{pos}\right)p\left(\mathrm{pos}\right)$$

The prevalence of infection is p(pos), also written *p*. Furthermore, p(neg)=1−p(pos). The final ingredient for a Bayesian analysis is the specification of priors, that reflects initial knowledge about the parameters. These are supplied as distributions, of which the variation is interpreted as uncertainty about the true value. Importantly, priors should be established independently, without using any of the data used for fitting. In this study, we set weakly informative priors on the unknowns, with a broad spread, expressing our prior ignorance on the true values. The full system of equations including priors is presented in Section A.1.

### 2.5. Sample Matrices

A distinct advantage of mixture models in a Bayesian setting is that stratifications and covariates can be included in the model in a natural way. In this study, for example, there is a distinction between the sample matrices ‘heart fluid’ and ‘serum’. This was modeled by adding a contribution to the mean when the matrix of observation *i* equals ‘serum’ (encoded by samplematrix[i]=2 in equations and model code). In the same spirit, the prevalence was estimated for each study: (1)$$\begin{array}{cc} f(x_{i}) & =\left(1-p_{i}\right)\times f\left(x_{i};\mu_{i}^{-},\sigma^{-}|neg\right)+p_{i}\times f\left(x_{i};\mu_{i}^{+},\sigma^{+}|pos\right), \end{array}$$(2)$$\begin{array}{cc} \mu_{i} & =\mu^{\mathrm{baseline}}+\mu_{samplematrix[i]}^{shift} \end{array}$$

Here, we write $\mu_{i}$ without superscript for simplicity, indicating that it applies to both the positive and negative component. The parameter $\mu^{\mathrm{baseline}}$ is the mean for serum, and $\mu^{\mathrm{shift}}$ is the shift needed to obtain the mean for the matrix ‘heart fluid’. Specifications for the priors were delegated to the Section A.1.

### 2.6. Censoring

OD values that were negative (due to measurement error) were added to the binary mixture model as ‘censored’ observations, which means that the observation is not counted as ‘a contribution to the likelihood due to an observation at some value’, but rather as ‘the observation that a value below a threshold was observed’. This still adds information to the model, and informs which parameters are to be estimated. Censored values were included in Equation (Equation 1), by replacing the probability density functions f() by their cumulative counterparts F(), and evaluated at a suitable low value representative of a value below which the measurements become uncertain. In this case, a logOD of −2 was chosen as the censoring point.

### 2.7. Prediction of Serological Status

The prevalence was estimated directly from the mixture model, but for other purposes (e.g., diagnosis or risk-factor analysis) an outcome per animal is often desired. After model parameters are established, the probability of being positive can be predicted per animal. The prediction of seropositivity for some given OD value x proceeds via Bayes’ theorem:(3)$$\begin{array}{cc} Pr(pos|x) & =\frac{Pr(x|pos)Pr(pos)}{Pr(x)} \end{array}$$(4)$$\begin{array}{cc} \; & =p\times \frac{f(x;\mu^{+},\sigma^{+})}{f(x)} \end{array}$$

Each of the parameters in this expression was estimated using software for Bayesian optimization, including full uncertainty posterior distribution as explained in the next section. Note that the individual results could be used to reconstruct the prevalence:(5)$$\hat{p}=\frac{1}{n}\sum_{i=1}^{n}Pr\left(\mathrm{pos}|x_{i}\right)$$

This indirect estimation of the prevalence $\hat{p}$ is not necessarily the same as the directly estimated prevalence *p*.

### 2.8. Model Accuracy

In order to assess model accuracy, we applied the defined cut-offs to the three data sets and compared the outcomes with RT-qPCR results. With RT-qPCR outcomes as a reference, standard model performance metrics can be calculated: accuracy (ACC: percentage true positive plus true negative), sensitivity, specificity, negative predictive value (NPV) and positive predictive value (PPV). For the mixture model, using Equation (Equation 3), we considered an animal with pr(pos|x)>0.5 as positive, and numerically solved *x*. In effect, this also sets a cut-off value $c_{\mathrm{mixture}}$ based on the mixture model only. Thus, even without any additional ’golden standard’ data, mixture models have an inherent theoretical sensitivity, given by the complement of the probability density function of the positive component in our mixture: (6)$$\begin{array}{cc} \mathrm{Se} & =Pr(\mathrm{test}\;\mathrm{scores}\;\mathrm{postivit}|\mathrm{the}\;\mathrm{animal}\;\mathrm{is}\;\mathrm{positive}) \end{array}$$(7)$$\begin{array}{cc} \; & =Pr(x>c_{\mathrm{mixture}}|\mathrm{positive}\;\mathrm{component}) \end{array}$$(8)$$\begin{array}{cc} & =1-F_{2}\left(c_{\mathrm{mixture}};\mu^{+},\sigma^{+}\right). \end{array}$$

Similar reasoning yields expressions for the remaining accuracy measures, which are presented in Section A.2.

### 2.9. Model Implementation and Checking

The model was implemented in the modeling language Stan [13], using the interface rstan [14] in R [15]. The source code is available as Section A.5. The output of a Bayesian model given the priors and likelihood is a posterior estimate of each parameter. This posterior estimate reflects a combination of our initial belief encoded in the prior, and the added information stemming from the data. The Stan language relies on simulation, and gives results using a large number of samples from the posterior distributions (in our case 6000 samples). These results were used for further processing (e.g., plotting, calculation of credible intervals, etc.) and to calculate probabilities at the individual level (Equation (Equation 5)). The model was run for 3000 iterations, with a warm-up phase of 1500 iterations, using four chains, and model convergence was checked by observing that chains have mixed (see Section A.3, Figure A1). Furthermore, the influence of the choice of prior was assessed (Section A.3), and it was found that priors did not influence the posterior outcomes.

## 3. Results

### 3.1. Plate-to-Plate Variation

Plate-to-plate variation in logOD values decreased after a correction (Figure 1). The effect of this correction on the distribution of logOD values for the samples is minor (Figure 2).

### 3.2. Serological Results and Cut-Off Value

For the calculation of the cut-off, the wild rat population was used. In the frequency distributions of logOD values per study, the existence of positive and negative components can be easily discerned, although the exact locations and spread are not immediately clear (Figure 2). Different cut-off values were calculated using the wild rat population. The most simple method, i.e., using the mean plus two or three times the standard deviation on the linear scale, resulted in cut-off values of −0.53 and −0.40 on the log scale (0.29 and 0.40 on the linear scale). Using the mean plus 2 or 3 times the standard deviation in log10-scale, the cut-off values obtained were −0.26 and 0.27, respectively, (0.55 and 1.86 on the linear scale), these are also indicated in Figure 2. In total, 35 OD values were negative, all in the wild-rat population. These negative values resulted from reference subtraction (measurement at 650 nm), and preclude the log-transform. Those OD values were removed from the cut-off value analysis.

### 3.3. Parameter Estimates from Mixture Modeling

The posterior estimates of the model parameters are presented in Table 2. The means of the components of each study are estimated very precisely, as are the standard deviations. The shift between heart fluid and serum is about 0.22 log10-units. Furthermore, the prevalence differs greatly between the studies. The posteriors are shown as full uncertainty distributions in Figure A2.

### 3.4. Model Performance

An insightful check for model fit is the comparison of the actual data with the fitted model. Figure 3 displays such a fit. The histograms represent the data, over all studies included. Overlayed are 50 transparent curves, each of which is plotted using a draw from the joint posterior distributions (i.e., paired draws; with dependence between uncertainty in the parameters).

Additionally, for all rats, the results of RT-qPCR are available, which were considered as a reference test for comparison. A range of common accuracy measures for ELISA results analyzed using different methods in comparison to RT-qPCR is shown in Table 3. The number of rats scored positive for each method is presented in Section A.3
Table A1. Figure 4a shows log10-transformed OD values for each study, split by RT-qPCR outcome. Individual animals are depicted as slightly jittered dots, and the cut-off lines give an indication of how animals are classified. For the binary mixture model, the actual probabilities of seropositivity for each rat were compared between RT-qPCR positive and RT-qPCR negative rats (Figure 4b). In particular, for the RT-qPCR negative rats, the great majority of rats have a probability of serological positivity near zero. The RT-qPCR positive rats show a slightly greater spread in the probability of serological positivity. In total, only five rats gave discordant results in the captive rat study, one rat had a 0.85 probability of being positive, but was RT-qPCR negative, and four rats were RT-qPCR positive but had probabilities ranging from zero to 0.3. In the feeder rat study, three RT-qPCR negative rats were positive in the mixture model, with probabilities close to one. Six rats were positive in the RT-qPCR, yet negative in the mixture model with probabilities up to 0.19. The mixture model concurred perfectly with the RT-qPCR outcomes for the wild rats.

## 4. Discussion

In this paper, we developed a binary mixture model in a Bayesian setting and used it on data of SEOV infection in rats. This methodology has a number of distinct advantages compared to the classical cut-off method. First, mixture models for serology are based on biologically plausible principles: lognormal distributions of groups of positive and negative individuals. This is in contrast to most cut-off methods where one hard threshold is postulated which is supposed to neatly divide the population in two. In some cut-off methods, this is accounted for by introducing a ‘doubtful’ category, however, this does not really solve the problem since a decision still needs to be made for the ‘doubtful’ category when prevalence is estimated. In binary mixture modeling, the prevalence is estimated directly from the data. Second, the outcome of a mixture model for individual samples is more informative than that of a hard classification based on a cut-off, since we have probabilities of positivity. Now, we do have an opportunity to assess our confidence in the infection status based on these probabilities. Third, when performing a Bayesian model fit, the uncertainty in the parameter estimates may be assessed. For the cut-off method, no such concept exists, only uncertainty due to the finite number of samples is taken into account, for example by bootstrapping the data. Finally, in the Bayesian mixture model, covariates can be included in a straightforward way, which we demonstrated by differentiating between the matrices ‘serum’ and ‘heart fluid’. Other extensions are possible, such as the inclusion of variables such as age, sex, country, species, etc.

Another appealing property of Bayesian modeling is that many data sources can be integrated in one model. We highlighted this by including three studies. Each study added information which enabled more precise estimates of the location and spread of the components. Data of future studies may also be incrementally added to the database, thereby continuously refining our estimates.

In this paper, we also performed a plate-to-plate correction following [12], but on the log10-transformed OD values. Another adaptation to the procedure was removing the constant term from the regression equation. This was based on the recognition that adding a constant to a log-normal distribution skews the shape, which hinders fitting a model based on detection of these shapes. Some commercial kits recommend using the sample-positive ratio, scaling each plate by the value of the positive control. This method does not take into account that the positive control also has some random fluctuation on top of the systematic bias. With a regression approach using several control sera, such fluctuations are averaged and less influential. In future work, it would be interesting to include the plate-to-plate variation directly in the Bayesian model.

As a negative population, we used a wild rat population which was considered to be negative, based on all-negative RT-qPCR results. Cut-off values were obtained in log10-scale and linear scale using the mean plus two or three times the standard deviation. The log-scale is a prerequisite for obtaining the specificities that are aimed for with a mean plus 2 or 3 standard deviations: 97.5% and 99.7%. Table 3 shows that indeed, the cut-off values based on log-transformed values come much closer to the desired specificities, albeit not perfectly. For the feeder rat study, this may in part be explained by the different matrix employed, which shifted the distributions to the right.

Furthermore, the binary mixture model did not always perform as well as theoretically predicted. Looking at the captive rats study, we find only a 71% sensitivity where 93% was expected (Table 3). In the study on feeder rats, we obtained 84% instead of the 97% expected. The ‘outlier points’ in Figure 4 give us a possible explanation, they seem to be truly RT-qPCR positive but serologically negative or vice versa. For example, in the study on feeder rats, there are 16 PCR negative rats, but three of those were scored positive by the mixture model, as well as by most cut-off methods. However, looking at the top three points, we can observe that those are clearly serologically positive. Taking this into account would raise the specificity from 13/16×100%≈81% to perfect specificity. Hence the theoretical mixture model performance could well be accurate while the supposed ’golden standard’ is lacking. Similar considerations also hold for the cut-off-based methods. The few rats for which serology and RT-qPCR results diverge could be rats that were recently infected before capture, or rats that did clear their SEOV infection. The finding of RT-qPCR negative and seropositive rats indicates that clearance of the virus may be possible in rare cases. However, these are exceptions, as most SEOV infections result in life-long infection in rats [16,17].

We found that the results for the different cut-off methods are very sensitive to the exact cut-off chosen. Naturally, this depends on the characteristics of the ELISA and the overlap between the two distributions. When a distinctive negative and positive distribution are present, the influence of the chosen cut-off is smaller than that compared to a situation with overlapping distributions, which is more common in adapted or in-house ELISAs for non-standard species or pathogens. In our SEOV ELISA, the result for a cut-off of the mean plus three standard deviations in linear scale works best (in terms of accuracy, sensitivity, and specificity), and it even slightly outperforms the binary mixture model. However, this was a lucky coincidence, as the cut-off choice is arbitrary, and without a ‘gold standard’ to compare with, one can never know which test is best in practice. Nonetheless, the added advantages of the binary mixture model outweigh this slight outperformance.

The sensitivities of the cut-off-based on three standard deviations are poor. Figure 3 shows why this occurs: the cut-off of the mean plus three standard deviations is ‘cutting off’ a large chunk of the positive component, mainly because of the shift to the right in the feeder rat study, which was based on heart fluid which is systematically higher than serum. Hence, a cut-off that works nicely in one situation may fail in other situation. In contrast, the binary mixture model has a stable performance.

There are some cases in which binary mixture models will not converge to stable parameter estimates. For both mixture models and cut-off models, the performance is very much dependent on the characteristics of the data. When the positive and negative populations overlap substantially, or when the prevalence is extremely small or large, both methods will have trouble giving meaningful results. However, crucially, when using a cut-off method, one can never know how inaccurate the obtained result is. Mixture models in contrast will either fail to converge, or give large credible intervals for parameter estimates, thereby signaling that the results may be unreliable.

Another problematic feature of the data could be that components are skewed and no longer resemble a proper log-normal distribution. Using mixture models, this may be solved by using non-Gaussian distributions [18,19]. A possible cause for this phenomenon is that at the extremes of the OD value spectrum, the optical densities no longer linearly depend on the antibody levels. A calibration curve could be included in the model to correct for this effect, or alternatively, the skewed parts of the model could be treated as censored data (the same technique as used in the current study for negative OD values). It could also be the case that the data is actually not log-normally distributed because it is a mixture of differing populations, age-groups, etc. Such a situation would require detailed study to find the root cause.

The use of binary mixture models for serological assays has been advocated since the 1990s [1,20]. Since then, several extensions to the basic Bayesian model have been proposed, such as the inclusion of dependence between multiple tests on single subjects [21], or integrated plate-to-plate variation using a hierarchical Bayesian set up [3]. It would be an interesting avenue of research to further develop Bayesian models to include all those aspects in one model.

Despite its advantages, the binary mixture modeling method has not gained widespread use in veterinary and clinical practice. Surely the complexity of the method as compared to a simple cut-off value plays a role in this. However, nowadays, researchers in life sciences are increasingly supported by (in-house) statistical or modeling expertise, to assist with data analyses. Furthermore, traditionally the focus has mostly been on the proper data collection and laboratory analyses, instead of the methods used for data-analysis.

It is our hope that this manuscript exposed the subject matter in such a way as to make it of practical use for those who are not expert statisticians. We aimed to develop our model code to be accessible for a broad audience. Having the source code available with this paper can perhaps facilitate the use of binary mixture models, and provide a basis to build further on. Having knowledge of the existence of more flexible and informative data-analysis methods will hopefully lead to the increased use of these.

## Figures and Tables

**Figure 1 viruses-13-01155-f001:**
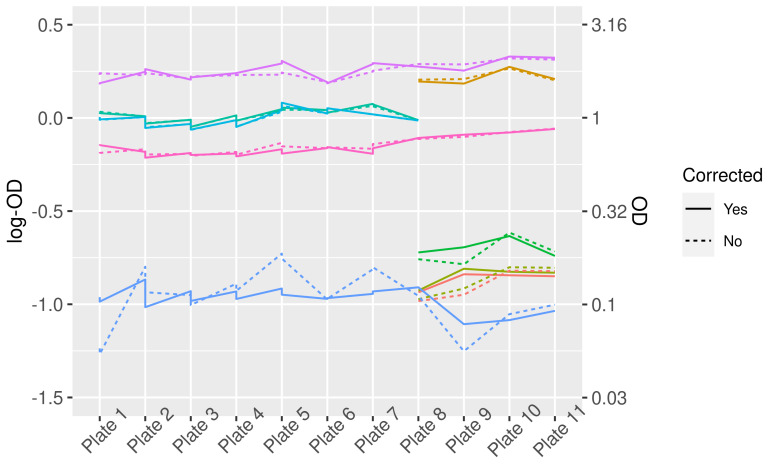
LogOD and OD values for control sera before and after correction for plate-to-plate variation. Different colors represent different control sera. Note that not all controls were present on each plate, therefore some lines do not cover the complete width of the graph.

**Figure 2 viruses-13-01155-f002:**
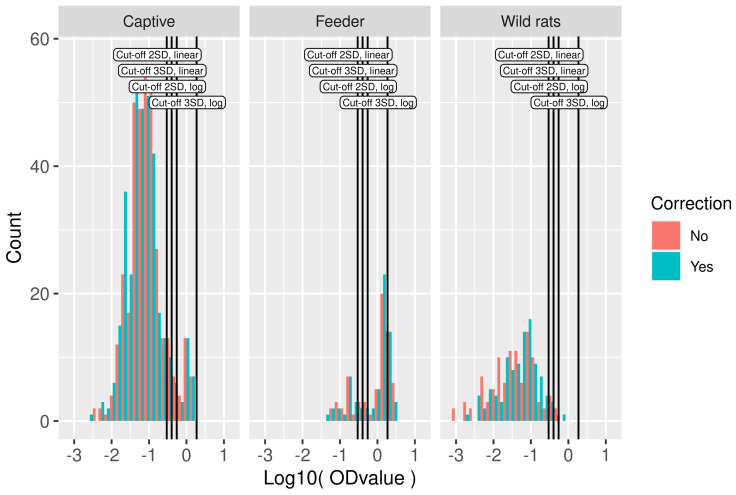
Serological data used in model fitting. Histogram of logOD values by study. Cut-off values based on the mean OD values in the wild rat dataset plus two or three standard deviations in log10-scale and linear scale are indicated by vertical lines. On the y axis, the count is presented, i.e., the number of rats with an OD value within the range of logOD values covered by the width of the bar.

**Figure 3 viruses-13-01155-f003:**
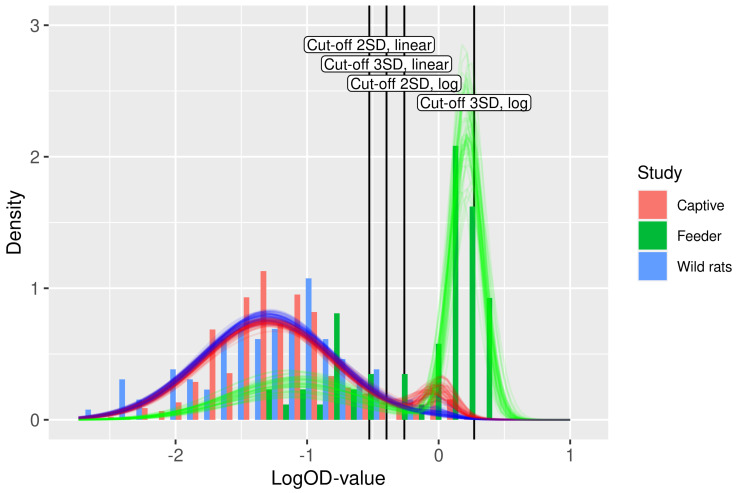
Fit of the binary mixture model (curves) to the ELISA logOD values (histograms). Each curve is plotted using a random draw (n=50) from the joint uncertainty distributions of the parameters in Table 2. Hence, the collection of curves gives an impression of the overall uncertainty in the model prediction. The color of the curves indicates the study. The four cut-off values are shown as vertical lines.

**Figure 4 viruses-13-01155-f004:**
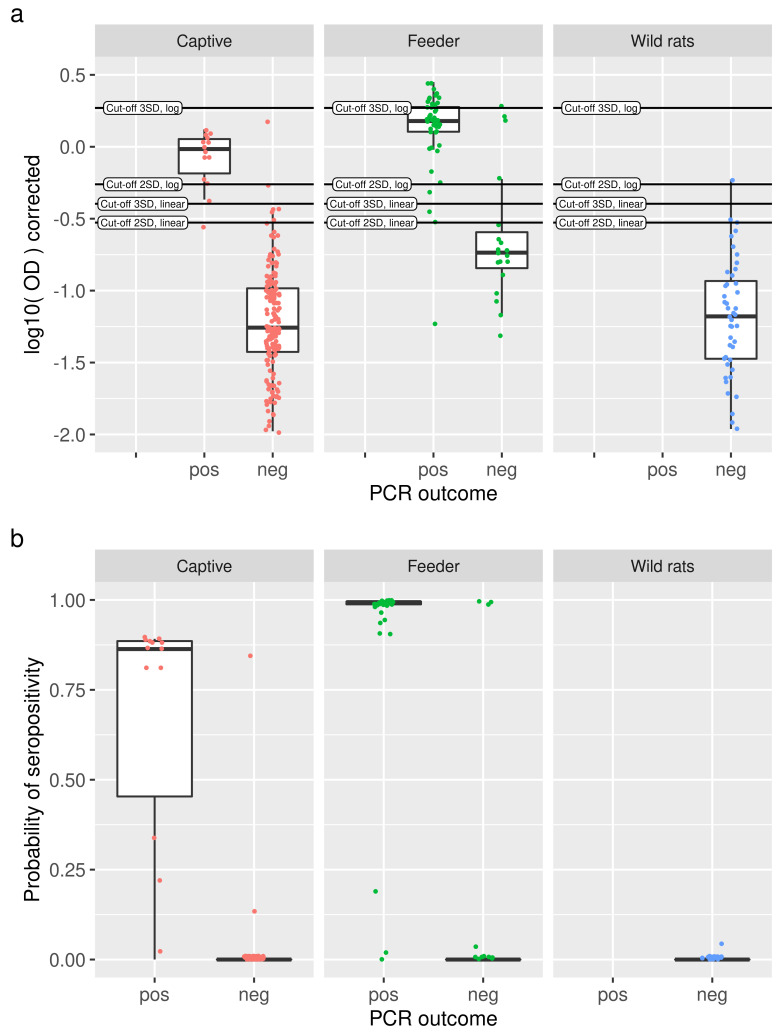
(**a**) For each study (panels), log-transformed OD values are shown as slightly jittered points with a boxplot, split into RT-qPRC positive and negative animals. Cut-off values are indicated as horizontal lines. (**b**) Comparison of predicted probabilities of seropositivity from the binary mixture models (dots, slightly jittered, with boxplot) with RT-qPCR outcomes (x axis) for the three studies under consideration (panels). With a few exceptions, high probabilities of seropositivity are observed for positive RT-qPCR results, and vice versa.

**Table 1 viruses-13-01155-t001:** Studies and the numbers of rats tested for each study, stratified by sample matrix. Note that in 2016, three additional rats were tested, but those were used as controls for plate-to-plate variation and not included in this table.

Sample Matrix	2018		2016
	Captive Rat Study	Wild Rat Study	Feeder Rat Study
heart fluid	175	69	2
serum	0	0	67

**Table 2 viruses-13-01155-t002:** Posterior estimates of the model parameters, with a 95% credible interval. Parameters $\mu^{\mathrm{shift}}$ and $p^{\mathrm{study}}$ are estimated at the study level, and are not different for the negative or positive components.

Parameter	Negative	Positive	Both
$\mu^{\mathrm{serum}}$	−1.30 (−1.35, −1.25)	−0.00 (−0.08, 0.06)	-
$\mu^{\mathrm{shift}}$	-	-	0.22 (0.15, 0.30)
σ	0.51 (0.47, 0.55)	0.12 (0.09, 0.15)	-
$p^{\mathrm{captive}}$	-	-	0.06 (0.03, 0.09)
$p^{\mathrm{feeder}}$	-	-	0.66 (0.54, 0.77)
$p^{\mathrm{wild rats}}$	-	-	0.01 (0.00, 0.03)

**Table 3 viruses-13-01155-t003:** Accuracy measures compared between the binary mixture model and the four cut-off choices (mean plus 2 × SD, or 3 × SD, each in both linear and log10 scale) with RT-qPCR. For the binary mixture model classification, the positivity of the rats was determined using the probability of positivity Pr(pos|x)>0.5. In some situations, the measures are not defined, e.g., in the captive rat study for cut-off values of the mean plus three standard deviations, the positive predictive value is not defined, since all rats were predicted negative. The theoretical binary mixture accuracy measures were not shown for the wild rat population, since they are not relevant for an all-negative population.

	Accuracy	Sensitivity	Specificity	PPV	NPV
Captive study					
Binary mixture P(pos) > 0.5	0.97	0.71	0.99	0.91	0.98
Binary mixture theoretical	0.98	0.93	0.99	0.81	1.00
Cutoff2	0.98	0.86	0.99	0.92	0.99
Cutoff2 linear	0.95	0.93	0.96	0.65	0.99
Cutoff3	-	0.00	1.00	-	0.92
Cutoff3 linear	0.98	0.93	0.99	0.87	0.99
Feeder study					
Binary mixture P(pos) > 0.5	0.72	1.00	0.00	0.72	-
Binary mixture theoretical	0.99	1.00	0.97	0.99	0.99
Cutoff2	0.72	1.00	0.00	0.72	-
Cutoff2 linear	0.72	1.00	0.00	0.72	-
Cutoff3	-	1.00	0.00	0.72	-
Cutoff3 linear	0.72	1.00	0.00	0.72	-
Wild rats					
Binary mixture P(pos) > 0.5	0.71	-	0.71	-	-
Cutoff2	0.70	-	0.70	-	-
Cutoff2 linear	0.67	-	0.67	-	-
Cutoff3	-	-	0.71	-	-
Cutoff3 linear	0.70	-	0.70	-	-

## Data Availability

The data presented in this study are openly available in https://github.com/arnoswart/MixtureModelSeoul.git (accessed on 29 April 2021).

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
