# Peer review of "Bayesian Binary Mixture Models as a Flexible Alternative to Cut-Off Analysis of ELISA Results, a Case Study of Seoul Orthohantavirus"

_viruses, 2021, doi:10.3390/v13061155_

Round 1

Reviewer 1 Report

Thank you very much for revising the manuscript and addressing the reviewers comments. In my opintion it is no eligible for publicaiton. However, there is still a small problem with the labelling in Figure 4 and in some figures in the Appendix. Please check and revise.

Author Response

Dear Reviewer,

In response to your request, and similar requests from another reviewer, we have reworked all of the figures to make them more readable.

Best wishes,

--Arno Swart

Reviewer 2 Report

Thanks for the revised version.

Author Response

Dear Reviewer,

In response requests from the other reviewers, we have reworked all of the figures to make them more readable. Thank you for your favourable assessment.

Best wishes,

--Arno Swart

Reviewer 3 Report

The revised version of the manuscript by Swart and colleagues has improved significantly. I believe my criticism on the last round included suggestions to improve the data presentation, and the authors have included proper figure legends for the revised version. This helps a lot. However, the authors must have extremely good eyesight because the labels on the figures are so tiny. I am willing to recommend accepting the manuscript after the authors have improved the figures as suggested:

-increase the fonts to at least double (e.g. Figure 2) or even triple (e.g. Figure 1)

-increase the size of Figure 1 to fit the page

-Figure 2 is somehow stretched, this looks unprofessional, please fix the fonts

-Figure 3, the text labels inside the figure (cutoffs) are readable, but the axis numbering is too small, and the axis legends are somehow distorted and could be larger.

-Figure 4, the cutoff labels are not shown fully. Figure is streched. Labels too small.

-Figure 1A, labels too small, unreadable

-Figure 2A Please reconsider the layout. The figure labels should all be under the legend, not separately under each figure. Labels too small, unreadable. 

For future reference, I recommend the authors improving the figure quality before initial submission. Nice figures give a good impression and -at least in my opinion- poorly constructed figures does the opposite.

Author Response

Dear Reviewer,

In response to your request, and similar requests from another reviewer, we have reworked all of the figures to make them more readable. We have also moved the captions of Fig A2 to one single caption.

Best wishes,

--Arno Swart

This manuscript is a resubmission of an earlier submission. The following is a list of the peer review reports and author responses from that submission.

Round 1

Reviewer 1 Report

The manuscript by Swart et al. describes a method for determining the cutoff for ELISA using Seoul orhtohantavirus infection of rats as the model. The manuscript is written in good-quality English; however, the manuscript does not in my opinion follow proper scientific presentation. Another concern own mine is that the manuscript does not necessarily fit the scope of Viruses very well because the focus of the manuscript is rather on developing a method for cutoff determination than diagnosing a virus infection. I think the manuscript would perhaps find better audience in a more diagnostics-orientated journal, but I think this would be up to the editor to decide.  Below are specific comments on the shortfalls of the manuscript:

-The introduction is as such fairly long, and does not contain adequate number of reference (the first reference appears on line 72, about 50 lines from the beginning!). Although ELISA is well-known technique, it is usually anyway spelled out. Normal ELISA does not necessarily measure antibody concentration (unless some sort of standard is used) but the result usually reflects the amount of antibodies. The detection in ELISA (Enzyme-linked immunosorbent assay) can be based on OD measurement when a chromogenic substrate is used, but the result can be read as e.g. luminescence too. In the context authors refer to i.e. measurement of antibodies (or serological testing), the result tells about presence/absence of antibodies and depending on the disease and antibody class measured that can mean ongoing or past infection. And ELISA is not only for measuring antibodies, one can measure e.g. antigen and all kinds of markers by ELISA. I suggest the authors read a bit about ELISA and include the references to the introduction.

-The ELISA used for measurements is not described well enough to judge the results. What is the antigen? What are the conjugates/secondary antibodies used? What is the substrate? What is the protocol? Is there a positive control? What kind of values does the positive control give? I would assume positive rats to have good amount of anti-nucleoprotein antibodies, and using e.g. TMB substrate I would expect the positives to be in the range of OD 2-4, assuming that a suitable dilution is used. Are the authors measuring IgG, IgA, or IgM? I know that the kit says IgG, but I guess the kit is originally for human samples anyway.

-Figure 1, what are the different colors? As such the graph tells very little. What are the different colors? Are the samples positive or negative controls? Why do not all the lines go through the x-axis? What does log-OD tell or mean? Is this figure in the right place (should it not appear following the first reference to it)?

-Results, line 215, to me it is not that clear that the logOD values decreased after the correction, at least based on Figure 1.  

-Figure 2, also this Figure is not clear enough to merit the short figure legend, it is quite hard to guess what the different lines mean. What does the y-axis represent? How many of the samples are positive with the different cutoffs?

-To me it is unclear why the cutoff values are negative, and why do they need to be presented as logOD? Showing the original OD values would give the reader better impression on the data.

-lines 227-229, the OD values should not be negative, show the original values. Was some sort of negative sample subtracted from the values? It seems by removing these values the cutoff for the test increases dramatically, which in turn reduces the sensitivity of the test dramatically.

-Figure 3, ???

-Using RT-qPCR for reference assumes that all infected animals remain persistently infected, which might not be the case

Also the rest of the manuscript would require a lot of revision, perhaps the authors could consult some of their peers at the institution to help in making the manuscript understandable for general audience? There might be interesting and useful findings in the manuscript, but it is very hard to judge the results because of the poor presentation of the data (the figures are not explained well enough, raw data not shown, the rationale for showing the results as logOD as opposed to regular OD values ranging from 0-4, etc.). As such, I do not think that the manuscript fulfills the criteria for a scientific report.

Reviewer 2 Report

viruses-1123339 

Bayesian Binary Mixture Models as a Flexible Alternative to Cut-off Analysis of ELISA Results, a Case Study of Seoul Orthohantavirus 

SUMMARY 

Serological assays, such as the ELISA, are popular tools for establishing the seroprevalence of various infectious diseases in humans and animals. In the ELISA, the optical density is measured and gives an indication of the antibody level. However, there is variability in optical density values for individuals that have been exposed to the pathogen of interest, as well as individuals that have not been exposed. In general, the distribution of values that can be expected for these two categories partly overlap. Often, a cut-off value is determined to decide which individuals to consider seropositive or seronegative. However, the classical cut-off approach based on a putative threshold ignores heterogeneity in immune response in the population and is thus not the optimal solution for analysis of serological data. A binary mixture model does include this heterogeneity, offers measures of uncertainty, and direct estimation of seroprevalence without the need for correction based on sensitivity and specificity. Further, a probability to be seropositive can be estimated for individual samples, and both continuous and categorical covariates (risk-factors) can be included in the analysis. Using ELISA results from rats tested for Seoul orthohantavirus, we compare the classical cut-off method with a binary mixture model set in a Bayesian framework. We show that it performs similarly or better than cut-off methods, by comparing with RT-qPCR results. We therefore recommend binary mixture models as an analysis tool over classical cut-off methods. Example code is included to facilitate practical use of binary mixture models in everyday practice.

COMMENTS

The paper deals with an interesting topic and the methods are sound. It may be worthwhile to examine prior sensitivity, however, which is an integral component in any Bayesian analysis.

Reviewer 3 Report

The manuscript describes a Bayesian binary mixure model which offers advantages to the widely used cut-off methods to analyse ELISA results. In general, the manuscript is well written, however lacks some references in the introduction section. Please check carefully, which statements need references and which reflect the authors opinions (in the whole manuscript). Additionally, I adressed some minor issues in the attached file.
